# Gait alterations in Parkinson's disease at the stage of hemiparkinsonism—A longitudinal study

Vladana Marković[1]*, Iva Stanković[1], Saša Radovanović[2], Igor Petrović[1], Milica Ječmenica Lukić[1], Nataša Dragašević Mišković[1], Marina Svetel[1], Vladimir Kostić[1]

**1** Clinic of Neurology, Faculty of Medicine, University of Belgrade, Belgrade, Serbia, **2** Institute for Medical Research, National Institute of Republic of Serbia, University of Belgrade, Belgrade, Serbia

* vladanaspica@gmail.com

**Data Availability Statement:** All relevant data files are available from the Kaggle database (https://www.kaggle.com/datasets/vladanamarkovic/gait-alterations-in-hemiparkinsonism).

## Abstract

### Background

Progressive gait impairment in Parkinson's disease (PD) leads to significant disability. Quantitative gait parameters analysis provides valuable information about fine gait alterations.

### Objectives

To analyse change of gait parameters in patients with early PD at the stage of hemiparkinsonism and after 1 year of follow up, taking into account clinical asymmetry.

### Methods

Consecutive early PD outpatients with strictly unilateral motor features underwent clinical and neuropsychological assessment at the study entry and after 1 year of follow up. Gait was assessed with GAITRite walkway using dual-task methodology. Spatiotemporal gait parameters (step time and length, swing time and double support time) and their coefficients of variation (CV), gait velocity and heel-to-heel base support were evaluated.

### Results

We included 42 PD patients with disease duration of 1.3 years (±1.13). Progression of motor and non-motor symptoms, without significant cognitive worsening, was observed after 1 year of follow up. Significant shortening of the swing time, prolongation of the double support and increase of their CVs were observed during all task conditions similarly for most parameters on symptomatic and asymptomatic bodysides, except for CV for the swing time under the combined task.

### Conclusion

Alterations of the swing time and double support time are already present even at the asymptomatic body side, and progress similarly, or even at faster pace, at this side, despite

**Funding:** The authors received no specific funding for this work.

**Competing interests:** The authors have declared that no competing interests exist.

dopaminergic treatment These parameters deserve further investigation in larger, prospective studies to address their potential to serve as markers of progression in interventional disease modifying trials with early PD patients.

## Introduction

Progressive gait disturbance is a major determinant of physical independence of patients with Parkinson's disease (PD), an important risk factor for falls [1–3], and a significant indicator of reduced health-related quality of life [4]. While distinctive pattern of shuffling gait in later stages of PD is well known since the first description of the disease, gait alterations in early PD are subtle. Analysis of quantitative data obtained using different types of sensors for gait recording provides valuable information on gait alterations that might prove useful as markers of disease progression. Common pattern of gait impairment specific to early PD includes reduced amplitude of arm swing, increased interlimb asymmetry, reduced smoothness of locomotion and increased cadence [1]. Even in prodromal stage of PD, asymptomatic carriers of LRRK2-G2019S mutation experienced gait alterations such as subclinical arm swing asymmetry and increased variability under dual task condition detected with wearable sensors [5]. However, these are not universal findings. An altered but symmetric gait pattern was observed in our de novo drug naïve PD patients with unilateral motor symptoms, possibly reflecting an activation of compensatory mechanisms for coping with balance problems at the earliest disease stage [6].

Data from prospective studies addressing natural history of gait impairment in early PD are scarce. In an incident PD cohort with patients in Hoehn and Yahr (H&Y) stages 1–3 a subtle deterioration of pace and rhythm was observed over 18 months of follow up despite optimal dopaminergic treatment [7], while asymmetry was revealed only after up to 6 years of follow up [8]. In another study, increased step number and step time variability were observed in early PD patients at normal-paced walking compared to healthy subjects over time [9]. However, none of these studies addressed PD patients with strictly unilateral motor features at study entry.

Our aim was to analyse changes in gait parameters in patients with early PD at the stage of hemiparkinsonism over 1 year of follow up, taking into account overt clinical asymmetry of motor features.

## Material and methods

### Subjects

Study included consecutive outpatients with PD diagnosis according to the Step 1 of the UK PD Society Brain Bank diagnostic criteria [10] at the H&Y stage 1 with strictly unilateral motor symptoms [11]. Patients were recruited at the Clinic of Neurology, Medical Faculty, University of Belgrade, Belgrade, from January 2012 to December 2013. Patients suffering from other concomitant neurological disorders, dementia, with history of psychosis requiring neuroleptic treatment, atypical and secondary parkinsonism or other diseases and conditions potentially affecting gait (for example, orthopaedic and rheumatology diseases) were not included in the study. Additionally, we excluded patients that fulfilled DSM-IV criteria for major depressive disorder.

All patients underwent clinical, neuropsychological and gait assessment at the study entry (t0) and after 1 year of follow up (t2).

The study was conducted in accordance with the Declaration of Helsinki and approved by the local Ethics Committee of Medical Faculty, University of Belgrade. All subjects gave written informed consent before the study inclusion.

## Clinical assessment

At study entry, data on the disease history were collected, including the first symptom noted by the patients. Disease severity was assessed with MDS-UPDRS, including non-motor and motor aspects of experiences of daily living [12], and levodopa equivalent daily dose was calculated [13]. The neuropsychological assessment, performed by experienced neuropsychologist, included the assessment of global cognitive functioning (mini-mental state examination–MMSE) and Addenbrooke's cognitive examination–revised (ACE-R) and cognitive domains: (a) memory—Rey auditory verbal learning test (RAVLT); (b) language—Boston naming test (BNT); (c) executive and working memory—phonemic and semantic fluency, digit ordering test; digit span backward; trail making test (TMT) and Stroop colour-word test; d) attention—digit span forward from Wechsler Adult Intelligence Scale and e) visuospatial—Hooper visual organization test [14–16]. We used Hamilton Depression Rating Scale (HDRS), Hamilton Anxiety Rating Scale (HARS) and Apathy Scale to evaluate severity of symptoms of depression, apathy and anxiety [17].

## Experimental protocol of gait recording

Gait was measured on GAITRite instrumented walkway with 5.5 m active area with built-in sensors reactive on pressure (CYR Systems, Havertown, PA) employing four test conditions. During gait recording patients were in their optimal "on" condition. First, patients walked 1.5 m before stepping on the walkway and finished their walk at least 1.5 m after the end of the walkway active surface, at their preferred walking speed ("basic gait condition"). Then, three additional test conditions were performed including a motor task (basic gait + motor task), a mental task (basic gait + mental task), and a combined motor and mental task (basic gait + motor task + mental task). The motor task consisted of walking with glass filled with water at subject's normal pace, aiming not to spill the water during the walk. The mental task comprised of concurrent walking and serial "7" subtraction, starting from randomly chosen numbers 90, 95, 100 or 105. Combined motor and mental task comprised of simultaneous walking while carrying the glass filled with water and performing subtraction as described above. Patients performed 6 passes, 3 times down the corridor and back for each test condition. They were walking for each test condition approximately 50 m (6 x 8–9 m), 200 m in total [18, 19]. Data from the activated sensors were collected. The GAITRite software enabled the calculation of temporo-spatial gait parameters, and the data were subsequently exported for further statistical analysis [20].

## Assessment of gait data

Spatiotemporal parameters of gait including velocity, heel-to-heel base support, step time, step length, swing time, double support time, and coefficients of variability (CVs) for step time, step length, swing time and double support time were analysed for all test conditions [21]. Averages of each parameter for each test condition were calculated separately for left and right leg with an exception of gait velocity and heel-to-heel base support that are measures derived from parameters from both legs. Then, as patients had unilateral motor features at t0, we stratified recordings into those obtained from symptomatic bodyside and those obtained from asymptomatic body side before the statistical analysis.

## Statistical analysis

We used independent samples T-test to compare gait parameters obtained from clinically symptomatic and asymptomatic bodyside at t0. The comparisons of relevant variables (clinical data and gait parameters) at t0 and t1 were performed using the Wilcoxon signed-rank test for nonparametric data and the Paired Sample T-Test for parametric data. Delta (Δ) values ((t1-t0/t0)*100) were calculated to analyse change (in percentages) of gait parameters over time at the symptomatic and the asymptomatic bodyside and were compared using T-test for independent samples (parametric) or Man Whitney U test (nonparametric data). To compare differences in Delta values across the task conditions Related-samples Friedman's Two-Way Analysis of variance by Ranks was used. Statistical analysis was performed using SPSS 20.0 (SPSS, Chicago, IL), and p values <0.05 were considered significant.

## Results

### Clinical assessment

Forty-two patients diagnosed with PD were tested at study entry (t0) and retested after 1 year (t1). Enrolled patients were more often male (61.9%) with mean age of 59 years (±9.75), disease duration of 1.5 years (±1.13) and levodopa equivalent daily dose of 131.6 mg (±190.72) at t0. Thirty-two (76.2%) PD patients reported an isolated initial symptom on the arm, 5 patients (11.9%) on the leg and 5 patients (11.9%) both on the arm and on the leg. Among 10 patients with the affection of the leg, 3 (7.1%) reported gait difficulties, while others reported tremor or stiffness. More severe symptoms in non-motor and motor aspects of experiences of daily living, as well as higher scores on motor examination and higher LED were observed in patients with PD at t1 compared to t0. A total of 17 patients (40.5%) progressed from H&Y stage 1 to 2. None progressed to more advanced motor stages or developed motor complications (neither dyskinesia, nor motor fluctuations), with the exception of painful dystonia (2 patients at t0 and 5 patients at t1). There was no statistically significant decline in global cognitive functioning, memory, learning, language, attention and visuospatial abilities (Table 1).

### Gait assessment

At t0, there were no differences in any of the gait parameters at the symptomatic bodyside compared to the asymptomatic bodyside suggesting a lack of asymmetry of quantitative gait parameters at this stage (see S1 Table).

Over time, no changes were found in the average gait velocity (basic gait 1.142 ± 0.033 vs. 1.149 ± 0.032 m/s; p = 0.758; motor task 1.097 ± 0.034 vs. 1.085 ± 0.037 m/s; p = 0.632; mental task 0.959 ± 0.036 vs. 0.969 ± 0.037 m/s; p = 0.647; combined task 0.912 ± 0.035 vs. 0.918 ± 0.034 m/s; p = 0.772), the average heel-to-heel base support (basic gait 9.6 ± 2.5 vs. 9.8 ± 2.8 cm; p = 0.441; motor task 9.4 ± 2.4 vs. 9.7 ± 3.1 cm; p = 0.441; mental task 10.1 ± 3.1 vs. 10.3 ± 3.2 cm; p = 0.441; combined task 9.7 ± 2.8 vs. 9.9 ± 3.2 cm; p = 0.441), and the step length and step time (for each bodyside) for all test conditions (Tables 2 and 3).

Significant shortening of the swing time and prolongation of the double support time at the initially symptomatic bodyside were observed during all test conditions at t1 compared to t0 (Table 2). The CV for swing time increased during basic gait condition, motor task and mental task conditions, while the CV for double support time increased during basic gait condition, motor task and combined task (Table 2). Similarly, spatial-temporal parameters recorded at initially asymptomatic bodyside suggested shortening of the swing time and increase of its CV during all test conditions at t1 compared to t0. The prolongation of the double support time

**Table 1. Baseline characteristics of patients and at one year follow up.**

|  | Baseline | Follow up | P |
|---|---|---|---|
| N | 42 | 42 |  |
| Gender (N, % male) | 26 (61,9%) | 26 (61,9%) |  |
| LEDD | 131.65 ± 190.72 | 340.73 ± 136.20 | <**0.001** |
| UPDRS I | 4.76 ± 3.47 | 6.49 ± 4.48 | **0.014** |
| UPDRS II | 5.49 ± 3.45 | 7.56 ± 4.19 | <**0.001** |
| UPDRS III | 15.24 ± 4.33 | 22.00 ± 8.04 | <**0.001** |
| UPDRS IV | 0.10 ± 0.49 | 0.17 ± 0.54 | 0.538 |
| UPDRS total | 25.58 ± 9.33 | 36.49 ± 13.84 | <**0.001** |
| Hoehn and Yahr stage | 1.21 ± 0.25 | 1.57 ± 0.41 | <**0.001** |
| HDRS | 5.02 ± 4.40 | 7.02 ± 5.85 | **0.022** |
| HARS | 4.71 ± 5.23 | 5.39 ± 4.83 | 0.434 |
| Apathy scale | 10.66 ± 8.05 | 11.20 ± 7.67 | 0.646 |
| ACE-R total | 91.83 ± 6.05 | 92.51 ± 5.71 | 0.412 |
| MMSE | 28.66 ± 1.20 | 28.32 ± 1.56 | 0.224 |
| RAVLT delayed recall | 7.683 ± 3.19 | 8.29 ± 2.74 | 0.206 |
| Digit forward | 8.32 ± 1.98 | 7.95 ± 1.91 | 0.217 |
| Digit backward | 6.38 ± 4.92 | 5.60 ± 1.77 | 0.314 |
| TMT | 47.51 ± 21.03 | 44.32 ± 16.99 | 0.307 |
| Stroop III | 42.76 ±19.84 | 41.08 ± 13.68 | 0.676 |
| Phonemic fluency | 26.89 ± 25.75 | 39.24 ± 10.52 | **0.003** |
| Categorial fluency | 18.81 ± 5.93 | 19.44 ± 5.63 | 0.462 |
| BNT total | 58.00 ± 2.66 | 58.05 ± 2.82 | 0.885 |
| Hooper | 19.70 ± 5.57 | 20.47 ± 4.71 | 0.526 |

Abbreviations: *LEDD* levodopa equivalent daily dose, UPDRS sponsored revision of the unified Parkinson's disease rating scale, *HDRS* Hamilton's scale of depression, *HARS* Hamilton's anxiety scale, *ACER* Addenbrooke's cognitive examination-revised, *MMSE* mini-mental state examination test, *RAVLT* Ray's auditory verbal learning test, *TMT* trail making test, *BNT* Boston's naming test, Hooper Hooper's test of visual organization.

was observed during all test conditions with an exception of combined task, while the CV of double support time was prolonged during each test condition with an exception of mental task condition (Table 3).

Delta values of change of these gait parameters were similar over all task conditions, e.g. additional tasks did not have an effect on further change of gait parameters. At self-paced walking (basic gait) we noticed shortening of swing time of 5.5% ± 6.2% at the symptomatic and 6.2% ± 6.9% at the asymptomatic bodyside, as well as prolongation of double support time of 9.2% ± 18.4% at the symptomatic and 12.5% ± 23.5% at the asymptomatic side (Fig 1, panel A and B). Meanwhile, their corresponding CVs increased markedly. During basic gait, swing time CV increased for 73.7% ±113.8% at the symptomatic, and for 101.1% ± 169.5% at the asymptomatic side, while double support time CV increased for 104.5% ± 243.9% at the symptomatic and 108.7% ± 157.4% at the asymptomatic side. The observed changes in the swing time, double support time and their corresponding CVs over 1 year follow up did not differ between initially symptomatic and asymptomatic bodysides during most test conditions. Significant difference between symptomatic and asymptomatic bodyside was noted only for the change in CV for the swing time under the combined task condition (54.2% ± 92.0% vs. 93.6% ± 111.5%, p = 0.046, respectively).

**Table 2. Gait parameters at baseline and after one year follow up of symptomatic leg at study entry.**

| Gait parameters | BASIC GAIT | | | MOTOR TASK | | | MENTAL TASK | | | COMBINED TASK | | |
|---|---|---|---|---|---|---|---|---|---|---|---|---|
| | Baseline | Follow up | p | Baseline | Follow up | p | Baseline | Follow up | p | Baseline | Follow up | p |
| Step time (s) | 0.56 ± 0.04 | 0.55 ± 0.05 | 0.289 | 0.56 ± 0.57 | 0.56 ± 0.07 | 0.543 | 0.62 ± 0.08 | 0.61 ± 0.10 | 0.56 | 0.63 ± 0.08 | 0.62 ± 096 | 0.513 |
| Step length (cm) | 61.92 ± 8.87 | 61.09 ± 8.58 | 0.338 | 59.59 ± 8.86 | 57.92 ± 9.29 | 0.089 | 55.94 ± 9.03 | 55.35 ± 9.09 | 0.49 | 54.00 ± 8.37 | 53.79 ± 8.18 | 0.826 |
| Swing time (s) | 0.39 ± 0.03 | 0.38 ± 0.03 | **<0.001** | 0.39 ± 0.03 | 0.37 ± 0.04 | **<0.001** | 0.42 ± 0.04 | 0.39 ± 0.06 | **<0.001** | 0.42 ± 0.04 | 0.39 ± 0.05 | **<0.001** |
| Double support time (s) | 0.32 ± 0.06 | 0.35 ± 0.08 | **0.008** | 0.34 ± 0.08 | 0.39 ± 0.12 | **0.001** | 0.40 ± 0.11 | 0.43 ± 0.14 | **0.04** | 0.42 ± 0.13 | 0.46 ± 0.14 | **0.039** |
| CV step time | 4.48 ± 2.29 | 6.16 ± 9.73 | 0.255 | 4.12 ± 1.59 | 6.95 ± 12.63 | 0.155 | 6.50 ± 3.02 | 8.90 ± 9.45 | 0.11 | 6.86 ± 4.49 | 8.80 ± 8.36 | 0.187 |
| CV step length | 4.44 ± 1.57 | 4.49 ± 6.01 | 0.873 | 4.69 ± 2.48 | 4.96 ± 3.05 | 0.513 | 7.38 ± 4.03 | 7.62 ± 6.77 | 0.84 | 7.22 ± 4.61 | 6.19 ± 2.77 | 0.131 |
| CV swing time | 5.12 ± 2.64 | 8.13 ± 5.23 | **0.001** | 5.29 ± 3.41 | 10.71 ± 7.79 | **<0.001** | 6.92 ± 3.65 | 10.55 ± 6.89 | **<0.001** | 7.48 ± 4.91 | 9.49 ± 5.50 | 0.082 |
| CV double support time | 9.60 ± 4.00 | 16.31 ± 13.69 | **0.006** | 8.85 ± 2.96 | 20.19 ± 27.54 | **0.01** | 13.73 ± 7.18 | 17.64 ± 15.23 | 0.15 | 13.67 ±7.61 | 25.13 ± 32.59 | **0.035** |

Values are shown as mean ± SD.

Abbreviation: CV –coefficient of variation.

**Table 3. Gait parameters at baseline and after one year follow up in asymptomatic leg at study entry.**

| Gait parameters | BASIC GAIT | | | MOTOR TASK | | | MENTAL TASK | | | COMBINED TASK | | |
|---|---|---|---|---|---|---|---|---|---|---|---|---|
| | Baseline | Follow up | p | Baseline | Follow up | p | Baseline | Follow up | p | Baseline | Follow up | p |
| Step time (s) | 0.55 ± 0.05 | 0.54 ± 0.05 | 0.393 | 0.55 ± 0.06 | 0.55 ± 0.07 | 0.91 | 0.60 ± 0.08 | 0.60 ± 0.11 | 0.73 | 0.61 ± 0.09 | 0.61 ± 0.11 | 0.713 |
| Step length (cm) | 62.65 ± 8.36 | 61.69 ± 8.13 | 0.241 | 60.05 ± 7.89 | 58.53 ± 8.71 | 0.091 | 57.04 ± 8.68 | 56.15 ± 8.83 | 0.3 | 54.68 ± 7.94 | 54.54 ± 7.75 | 0.874 |
| Swing time (s) | 0.39 ± 0.03 | 0.37 ± 0.03 | <**0.001** | 0.39 ± 0.03 | 0.36 ± 0.04 | <**0.001** | 0.41 ± 0.03 | 0.38 ± 0.05 | <**0.001** | 0.40 ± 0.04 | 0.38 ± 0.05 | **0.002** |
| Double support time (s) | 0.32 ± 0.06 | 0.36 ± 0.11 | **0.005** | 0.34 ± 0.08 | 0.38 ± 0.12 | **0.002** | 0.39 ± 0.11 | 0.44 ± 0.16 | **0.02** | 0.42 ± 0.13 | 0.45 ± 0.14 | 0.125 |
| CV step time | 4.54 ± 2.05 | 6.59 ± 10.41 | 0.194 | 3.98 ± 1.38 | 6.76 ± 9.24 | 0.064 | 7.45 ± 6.37 | 9.17 ± 10.90 | 0.38 | 6.59 ± 3.81 | 7.93 ± 10.03 | 0.421 |
| CV step length | 4.26 ± 1.86 | 4.44 ± 1.88 | 0.63 | 4.42 ± 1.99 | 4.96 ± 2.91 | 0.158 | 6.76 ± 3.06 | 7.82 ± 10.80 | 0.53 | 6.49 ± 2.74 | 5.91 ± 2.89 | 0.122 |
| CV Swing time | 5.26 ± 3.27 | 9.13 ± 6.64 | **0.001** | 4.75 ± 1.92 | 10.62 ± 7.74 | <**0.001** | 6.47 ± 2.66 | 11.14 ± 6.51 | <**0.001** | 6.41 ± 3.13 | 11.22 ± 5.93 | <**0.001** |
| CV Double support time | 9.93 ± 4.19 | 20.64 ± 22.84 | **0.003** | 9.01 ± 3.57 | 20.15 ± 28.81 | **0.018** | 12.63 ± 5.06 | 19.56 ± 26.70 | 0.11 | 13.16 ± 7.21 | 18.26 ± 11.02 | **0.004** |

Values are shown as mean ± SD.

Abbreviation: CV—coefficient of variation.

A. Change of swing time (%) and the respective coefficient of variation over 1 year

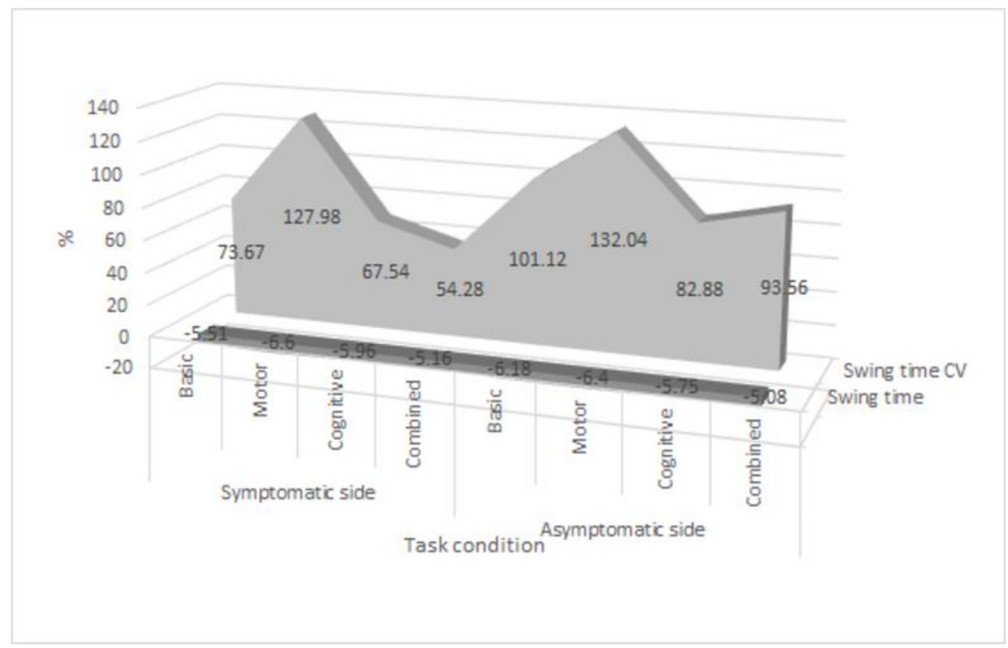

B. Change of double support time (%) and the respective coefficient of variation over 1 year

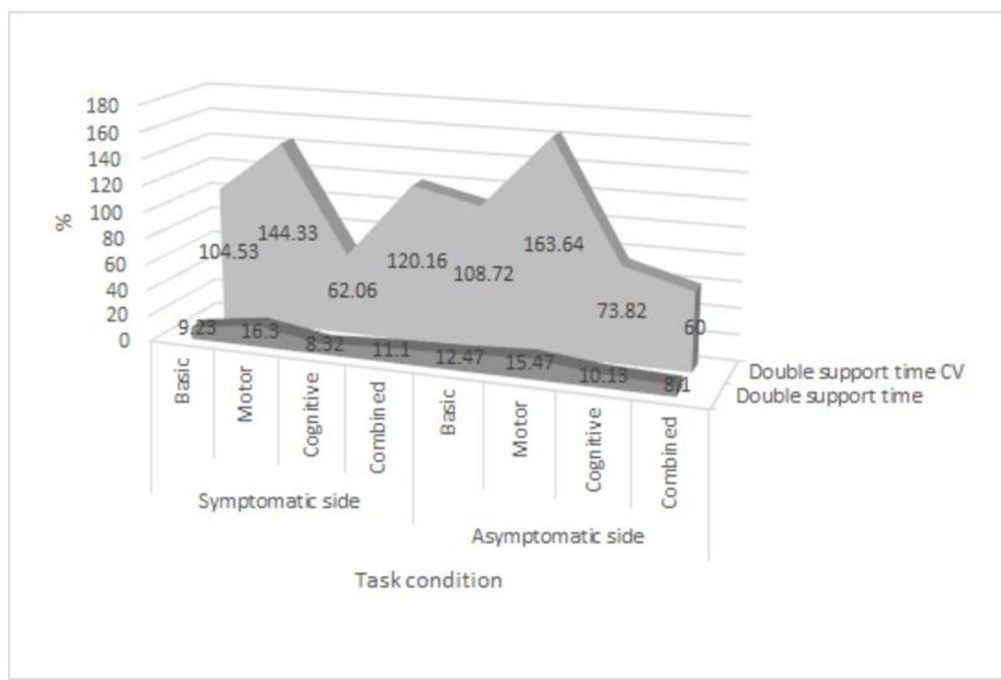

**Fig 1. Swing time and double support time and their respective coefficient of variations.** A. Change of swing time (%) and the respective coefficient of variation over 1 year. B. Change of double support time (%) and the respective coefficient of variation over 1 year.

## Discussion

First five years of PD treatment are traditionally considered as a "honey moon" period when excellent control of symptoms can be easily achieved [21]. However, our patients displayed significant motor worsening as well as more severe depression symptoms over 1 year follow up despite optimal dopaminergic treatment without complications of dopaminergic therapy. Extensive neuropsychological battery did not show significant cognitive decline in any cognitive domains. Lack of asymmetry of gait parameters at study entry, despite overt clinical asymmetry, suggests preclinical gait changes that can be attributed to symptom onset mostly at upper extremity in majority of our patients (more than ¾ of participants).

Alterations of many gait parameters in mild to moderate PD were reported in comparison to healthy controls, including lower cadence and longer stride time, as well as shorter swing time [22]. Patients with early PD (H&Y stage 1–2) exhibited significantly slower walking speed and shorter stride length, which significantly correlated with balance parameters [23]. In contrast, our study did not show worsening of the most robust parameters, such as velocity, heel-to-heel base support, step length and step time, possibly because patients were assessed in the earlier disease stage (unilaterally affected patients with disease duration 1.5 years (±1.13), suggesting that these parameters become altered later over the disease course.

Step length shortening during a cognitive task was an independent predictor of future executive/attention decline in early PD [24]. In our study, absence of shortening of step length and overall similar change of parameters during different task conditions is presumably due to preserved cognitive strategies in our early PD patients. Thus, dual task methodology seems unable to elicit more robust gait changes compared those observed during normal paced gait in early PD at the stage of hemiparkinsonism. Rather, analysing segments of gait cycle revealed shortening of swing time and prolongation of double support time, accompanied by significant increase in variability of these parameters in our patients. These changes over time were observed both on symptomatic and on the asymptomatic bodyside at study entry, whereas swing time variability under dual task condition exhibited even greater increase at the initially asymptomatic bodyside, suggesting possible breakdown of compensatory mechanisms as patients progress to the bilateral stage of the disease.

Interestingly, in a group of de novo PD patients (H&Y stage 1.88 ± 0.40), where decrement in stride length was shown, levodopa trial shortened stride time, resulting in the increment of cadence, presumably as compensatory mechanism for slow walking [25]. Our study included patients under optimal dopaminergic treatment, but revealed that gait impairment progressed despite dopaminergic therapy, suggesting that it is, at least to some extent, levodopa resistant. Alternatively, "ceiling effect" of levodopa, such early in disease course, can be suspected. Levodopa-resistant gait characteristics were assessed previously in early PD [26]. Low baseline CSF Aβ42, and to a lesser extent Aβ40, predicted worsening of gait variability in the first 3 years following diagnosis of PD suggesting amyloid pathology as possible contributing underlying mechanism [26].

Finally, our study has limitations. While in part our findings could be attributed to depression which was shown to affect gait parameters, in particular swing time variability [27], we excluded patients with major depression, and, while depressive symptoms have worsened over follow up, overall Hamilton depression rating scale score remained well below proposed threshold for depression (5.02 ± 4.40 vs. 7.02 ± 5.85; p = 0.022). Additionally, we did not take into account weight changes nor level of exercise as potential factors that could had had some impact on our findings. Our study sample was relatively small and relevance of these results needs to be confirmed in larger studies with longer follow up. In particular, it would be interesting to obtain the "real world" findings from wearable sensors in PD patients at earliest

disease stage. Nevertheless, taking into account affected body side during data analysis provides an insight into fine details of gait alterations occurring at the stage of hemiparkinsonism.

In conclusion, presented data suggest that changes of the swing time and double support time are already present very early at the disease course, while large majority of patients still do not complain of them, and even at the asymptomatic body side. These alterations progress similarly, or even at faster pace, at initially unaffected bodyside, despite optimal dopaminergic treatment. These parameters might prove useful as biomarkers of disease progression in interventional disease modifying trials with patients with early PD.

## Supporting information

**S1 Table. Gait parameters at baseline of symptomatic leg vs. asymptomatic leg at study entry.**
(DOCX)

## Acknowledgments

This work has been supported by the Ministry of Education, Science, and Technological Development, Republic of Serbia.

## Author Contributions

**Conceptualization:** Vladana Marković, Iva Stanković, Saša Radovanović, Igor Petrović, Vladimir Kostić.

**Formal analysis:** Vladana Marković, Iva Stanković, Saša Radovanović, Milica Ječmenica Lukić.

**Investigation:** Igor Petrović, Nataša Dragašević Mišković.

**Methodology:** Saša Radovanović, Milica Ječmenica Lukić.

**Project administration:** Saša Radovanović.

**Supervision:** Vladana Marković, Nataša Dragašević Mišković, Marina Svetel, Vladimir Kostić.

**Validation:** Vladana Marković.

**Writing – original draft:** Vladana Marković, Iva Stanković, Saša Radovanović, Igor Petrović.

**Writing – review & editing:** Marina Svetel, Vladimir Kostić.

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
