## [Decision Letter · Decision Letter 0]

21 Mar 2022

PONE-D-22-04031Gait alterations in Parkinson’s disease at the stage of hemiparkinsonism – a longitudinal studyPLOS ONE

Dear Dr. Markovic,

Thank you for submitting your manuscript to PLOS ONE. After careful consideration, we feel that it has merit but does not fully meet PLOS ONE’s publication criteria as it currently stands. Therefore, we invite you to submit a revised version of the manuscript that addresses the points raised during the review process (please, see below).

We look forward to receiving your revised manuscript.

Kind regards,

Federica Provini

Academic Editor

PLOS ONE

Journal Requirements:

Reviewers' comments:

Have the authors made all data underlying the findings in their manuscript fully available?

Reviewer #1: No

The authors have investigated the progression of various gait impairment parameters from baseline (t0) to 1 year (t1) in a sample of 42 Parkinson’s Disease (PD) patients with early onset PD and strictly unilateral parkinsonian features at t0 (H&Y=1). Gait assessment was performed using the GAITRite walkway, while patients were given specific predefined instructions to perform specific tasks (basic walking, motor task, mental task, combination). This is an original and interesting topic, with authors presenting original research, whose results has not been presented elsewhere. Authors have described experiments and statistics in sufficient detail and conclusions support the presented results. The article is well-presented and well-written and highlights the differences with previously published similar research.

I would like to make the following points:

1) In line 146, it is mentioned that: “Progression of motor and non-motor symptoms, decreased functioning in activities of daily living and higher LED were observed in patients with PD at t1 compared to t0”: how were the clinical assessment of non-motor symptoms and ADL performed? From what is written in the methodology section, I assume this information was extracted from the UDPRS I & II, but could the authors please be more specific in the Methodology?

2) In lines 148-150, it is mentioned that “None progressed to more advanced motor stages or developed motor complications (neither dyskinesia, nor motor fluctuations)”, which means that UDPRS IV should be zero (0) for t0 and t1. However, in table 1, it seems that UDPRS IV is not zero neither in t0 nor in t1, could the authors please elaborate a little more on this?

3) I would suggest adding the measurement units wherever applicable e.g. in table 2 presenting step time or step length.

4) I would suggest adding a citation for the GAITRite walkway in order to certify its use as a validated tool for gait assessment.

5) In line 209, could the authors please be a little more detailed on what does psychiatric worsening mean? Are they referring to the worsening in depressive symptoms?

6) In the discussion part, the authors mention that despite what has been observed in other studies assessing early PD patients, in this current study no worsening has been shown in gait parameters like velocity or stride length (lines 215-218). Would the authors consider the level of exercise or changes in weight between t0 and t1 as factor that might have contributed to these results?

7) In line 256, it is mentioned that patients did not complain of gait impairment, even when parkinsonism became bilateral at t1 (H&Y=2): was there any particular question assessing the subjective patient’s impression on gait impairment or an objective measurement that led the authors to this conclusion? Maybe the relevant questions in UDPRS about gait impairment?

8) In line 232, it is mentioned that “swing time variability under dual task condition exhibited even greater increase at the initially asymptomatic bodyside”, while in line 257, it is mentioned that “These alterations progress similarly, or even at faster pace, at initially affected bodyside”: could the authors please specify whether the increase was greater in the symptomatic or asymptomatic side during the combined task?

9) In line 181, it is mentioned that “spatial-temporal parameters recorded at initially asymptomatic bodyside suggested shortening of the swing time and its CV during all test conditions at t1 compared to t0”: did the authors mean that CV was increased and not shortened?

---

## [Author Response · Author response to Decision Letter 0]

6 May 2022

Dear Prof. Provini,

In the separate file, "Response to Reviewers" find enclosed the list of changes against each point that has been raised by the Reviewer and Editor, as requested. We hope our manuscript is now improved. 

Sincerely, 

Vladana Markovic

---

## [Editor Report · Decision Letter 1]

30 May 2022

Gait alterations in Parkinson’s disease at the stage of hemiparkinsonism – a longitudinal study

PONE-D-22-04031R1

Dear Dr. Markovic,

We’re pleased to inform you that your manuscript has been judged scientifically suitable for publication and will be formally accepted for publication once it meets all outstanding technical requirements.

Kind regards,

Federica Provini

Academic Editor

PLOS ONE

---

## [Editor Report · Acceptance letter]

12 Jul 2022

PONE-D-22-04031R1 

Gait alterations in Parkinson’s disease at the stage of hemiparkinsonism – a longitudinal study 

Dear Dr. Marković:

I'm pleased to inform you that your manuscript has been deemed suitable for publication in PLOS ONE. Congratulations! Your manuscript is now with our production department. 

Kind regards, 

on behalf of

Dr. Federica Provini 

Academic Editor

PLOS ONE